# Functional and structural analyses of IMP-27 metallo-β-lactamase: evolution of IMP-type enzymes to overcome Zn(II) deprivation

Yoshiki Kato,[1,2,3] Toshio Yamaguchi,[4] Haruka Nakagawa-Kamura,[5,6] Yoshikazu Ishii,[5,6] Akiko Shimizu-Ibuka[1,3]

**ABSTRACT** IMP-type metallo-β-lactamases are di-Zn(II) enzymes that can inactivate a wide range of bicyclic β-lactam agents used in clinical practice. IMP-27 shares 82% amino acid sequence identity with IMP-1, the first IMP-type enzyme identified. Herein, we conducted structural determination, kinetic, and chelating agent resistance analyses of IMP-27. Once determined, IMP-27 was then compared to its mutant, namely, G262S, and IMP-1. Crystallographic structural analysis of IMP-27 showed an overall structure comparable to that of IMP-1 and other IMP-type enzymes; the positions of the zinc (Zn) ions varied across enzymes. Kinetic analysis showed that IMP-27 had lower catalytic efficiency against penicillins, ceftazidime, cephalexin, and imipenem than IMP-1; however, it had higher affinity and catalytic efficiency against meropenem, especially in the presence of Zn(II). This suggests that the catalytic site of IMP-27 is optimized to hydrolyze meropenem during molecular evolution at the expense of catalytic efficiency against penicillins. However, Zn(II) content analysis after EDTA treatment revealed no significant difference between enzymes. Moreover, analysis of IMP-27 mutants indicated that the differences in kinetic properties and chelator resistance between IMP-1 and IMP-27 were mainly due to an amino acid substitution at position 262.

**IMPORTANCE** The residue at position 262 has been reported as a key determinant of substrate specificity in IMP-type enzymes. Among more than 80 IMP-type metallo-β-lactamase (MBL) variants, IMP-27 was the first reported IMP-type MBL isolated from *Proteus mirabilis*. This enzyme has a glycine residue at position 262, which is occupied by serine in IMP-1. Compared with IMP-1, IMP-27 had a significantly higher affinity and catalytic efficiency against meropenem and improved metal-binding capacity, maintaining its activity under Zn(II)-limited conditions better than IMP-1. The analysis of the IMP-27 mutants indicated that differences between IMP-27 and IMP-1 were mainly due to an amino acid substitution at position 262. In the case of IMP-27, the G262S mutation optimized the catalytic site of IMP-27 for meropenem hydrolysis, at the expense of catalytic efficiency against penicillins.

**KEYWORDS** antibiotic resistance, metallo-β-lactamase, X-ray crystallography

Antibiotic resistance is a major global public health concern. β-Lactam antibiotics are commonly used to treat pathogenic bacterial infections; however, their clinical use is often compromised by antibiotic resistance in bacteria (1). The major resistance mechanism against β-lactam antibiotics is the production of β-lactamase (EC 3.5.2.6), especially in Gram-negative bacteria (2). β-Lactamases consist of two groups of enzymes with different catalytic mechanisms: serine β-lactamases (SBLs), which use serine as a catalytic residue, and metallo-β-lactamases (MBLs), which require one or more Zn ion(s) for catalysis (3). β-Lactamases are classified into four classes (A–D) based on their amino

Address correspondence to Akiko Shimizu-Ibuka, ibuka@kanagawa-u.ac.jp.

The authors declare no conflict of interest.

See the funding table on p. 13.

acid sequences. Class A, C, and D β-lactamases are SBLs, while class B enzymes are MBLs (4). MBLs have lower catalytic efficiencies than SBLs (5); however, they are considered threats because of their wide-spectrum activity, exhibiting hydrolytic activity toward carbapenems, a class of β-lactam antibiotics generally stable against SBLs. They are also of particular concern because of the unavailability of clinically useful inhibitors, although boronate-based inhibitors have recently been developed and are currently undergoing clinical trials (6–9). Understanding the molecular properties of MBLs is necessary to overcome clinical infections caused by MBL-producing bacteria.

MBLs are divided into three subclasses, B1, B2, and B3, based on similarities in their amino acid sequences (10, 11). Subclasses B1 and B3 are di-Zn(II) MBLs, with the first Zn ion (Zn1) bound to the 3H site composed of three histidine residues (His116, His118, and His196), which are conserved in both subgroups (12). The second Zn ion (Zn2) binds to the DCH site, composed of Asp120, Cys221, and His263 in the B1 subgroup; cysteine is replaced with a histidine residue in B3 enzymes. Clinically significant MBLs, such as NDM-, VIM-, and IMP-type enzymes, are classified into the B1 subgroup.

The IMP-1-encoding gene was first isolated from *Serratia marcescens* in Japan in 1991; IMP-type enzymes remain predominant in Southeast Asia (13–15). IMP-1 was also isolated from *Enterobacterales* and *Pseudomonas aeruginosa* (16). There are currently >80 variants in the IMP group, with amino acid sequence identities ranging from 79.3% to 99.1% (17, 18). These enzymes have broad substrate specificity and high affinity for cephalosporins and carbapenems. IMP-27 is the first reported IMP-type MBL isolated from *Proteus mirabilis*. Two *P. mirabilis* isolates producing IMP-27 were collected from two patients in two different states in the United States. Both isolates harbored $bla_{IMP-27}$ as part of the first gene cassette in a class two integron, whereas most genes encoding IMP enzymes were found in class a or class three integrons (19). IMP-27 and IMP-1 differ by 42 amino acid residues and share 82% identity in their primary structures. IMP-27 contains a glycine residue at position 262, which is occupied by serine in IMP-1. The residue at this position has been known to be one of the key determinants of the kinetic properties of IMP-type enzymes (20).

Compared with IMP-1, IMP-27 has a lower catalytic efficiency, whereas it is more resistant to chelating agents, such as EDTA and 2,6-dipicolinic acid (DPA) (19). This study investigated IMP-27, its mutant G262S, and IMP-1 to elucidate how enzymatic properties change during the molecular evolution of IMP-type enzymes. Specifically, we performed structural determination analyses, kinetic analyses in the presence or absence of excessive Zn(II), direct measurements of metal content using inductively coupled plasma atomic emission spectroscopy (ICP-AES), and chelating agent resistance analyses to evaluate these changes.

## RESULTS

### Crystal structure of IMP-27

We analyzed the three-dimensional structure of IMP-27 at a 1.7 Å resolution (Table S1). The overall structure of IMP-27 was comparable to previously reported IMP-type enzymes (Fig. 1A). The structure of the active site in IMP-27 overlapped well with those of other IMP-type enzymes, including the main chain of the residue at position 262. However, the position of the His263 side chain, a Zn2 ligand, was slightly different from the other residues of the Zn ligand among IMP-type enzymes, including IMP-27 and IMP-1 (Fig. 1B). Two Zn(II) ions, Zn1 and Zn2, coordinated to the active center of IMP-27 similar to those observed in the crystal structures of other IMP-type enzymes. While the position of Zn1 overlapped well, the position of Zn2 varied between IMP-27 and most other enzymes, except IMP-6 (Fig. 1B; Table S2). Two IMP-27 molecules (A and B) in an asymmetric unit made protein–protein contact through a loop region containing β-strand B8, with a buried interface area of 358.0 Å$^2$. Each molecule also contacted its neighboring symmetric molecule through a region containing the L3 loop, with a buried interface area of 582.2 Å$^2$. The termini of the two β-strands B2 and B3 were also present in this region, forming a short four-stranded β-sheet through this contact (Fig. 1C).

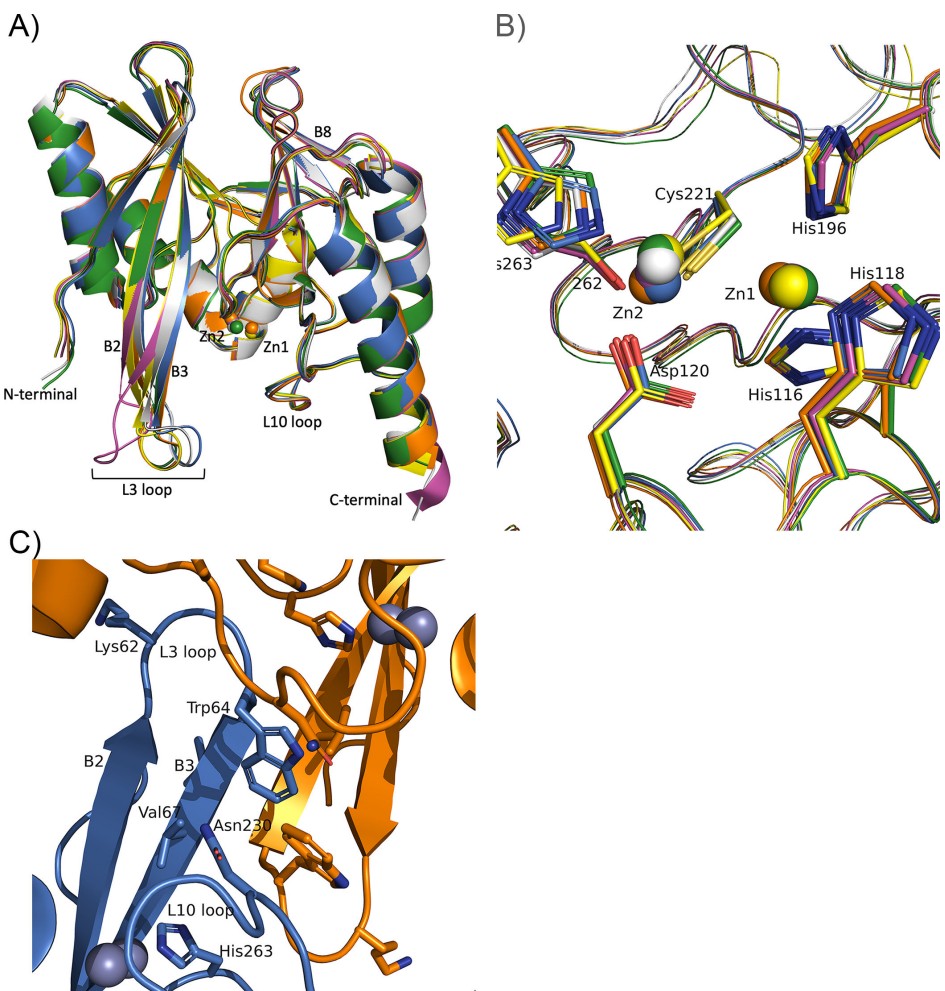

**FIG 1** Structure of IMP-27 compared with other IMP-type enzymes. (A) Overall structure and (B) active center of IMP-27 superimposed with those of other IMP enzymes whose structures are known. In panels (A) and (B), the structure of IMP-27 is shown in orange, IMP-1 (PDB ID: 5Y5B) in white, IMP-2 (PDB ID: 4UBQ) in yellow, IMP-6 (PDB ID: 6LVJ) in blue, IMP-13 (PDB ID: 6R79) in magenta, and IMP-18 (PDB ID: 5B3R) in green. (C) Crystal packing of IMP-27. One molecule and its symmetrically related neighbor are shown in blue and orange, respectively.

## Kinetic analyses

We performed kinetic analyses of IMP-27 and IMP-1 in the presence and absence of Zn(II) (Table 1). In the presence or absence of Zn(II), IMP-27 showed significantly lower catalytic efficiency against penicillins and cephalexin by more than one order of magnitude compared with IMP-1. For cephalexin, IMP-27 had smaller $k_{cat}$ and $K_m$ values than IMP-1. For meropenem, the $K_m$ of IMP-27 was smaller than that of IMP-1 by one order of magnitude, resulting in an approximately fivefold higher catalytic efficiency. For imipenem, the $K_m$ of IMP-27 was larger than that of IMP-1, resulting in a lower $k_{cat}/K_m$ of approximately one order of magnitude.

In IMP-27, adding 50 µM Zn(II) to the reaction solution significantly increased the catalytic turnover rate toward meropenem by fourfold. It also caused a slight increase in the affinity toward the tested cephalosporins, except for cephalexin, increasing the $k_{cat}/K_m$. The addition of Zn(II) caused no drastic changes in IMP-1; however, a slight increase in $K_m$ for penicillins and some cephalosporins was observed. Regarding imipenem, the addition of Zn(II) increased the $k_{cat}/K_m$ value by two-fold in both IMP-27 and IMP-1.

**TABLE 1** Kinetic parameters of IMP-1, IMP-27, and IMP-27G262S[a]

| Substrates | Zn (µM) | IMP-1 | | | IMP-27 | | | IMP-27G262S | | |
|---|---|---|---|---|---|---|---|---|---|---|
| | | $k_{cat}$ (s$^{-1}$) | $K_m$ (µM) | $k_{cat}/K_m$ (s$^{-1}$·µM$^{-1}$) | $k_{cat}$ (s$^{-1}$) | $K_m$ (µM) | $k_{cat}/K_m$ (s$^{-1}$·µM$^{-1}$) | $k_{cat}$ (s$^{-1}$) | $K_m$ (µM) | $k_{cat}/K_m$ (s$^{-1}$·µM$^{-1}$) |
| Ampicillin | 0 | 37.6 ± 3.1 | 41.8 ± 7.6 | 0.899 | >7.25 | >250 | 0.0288 | 8.25 ± 1.56 | 27.0 ± 7.2 | 0.306 |
| | 50 | **ND** | **ND** | **0.707** | **ND** | **ND** | **0.0307** | **ND** | **ND** | **0.481** |
| Benzylpenicillin | 0 | 367 ± 38 | 144 ± 6 | 2.56 | 20.3 ± 2.2 | 218 ± 28 | 0.0932 | 90.5 ± 10.8 | 48.2 ± 4.8 | 1.88 |
| | 50 | **597 ± 71** | **314 ± 25** | **1.90** | **27.2 ± 3.1** | **325 ± 50** | **0.0837** | **185 ± 11** | **71.4 ± 10.7** | **2.59** |
| Cephalothin | 0 | 82.5 ± 5.9 | 1.62 ± 0.21 | 51.1 | 290 ± 39 | 9.23 ± 0.88 | 31.4 | 35.2 ± 3.8 | 0.590 ± 0.096[b] | 59.7 |
| | 50 | **50.7 ± 8.7** | **2.51 ± 0.41** | **20.2** | **381 ± 39** | **6.56 ± 0.71** | **58.1** | **26.0 ± 1.9** | **0.346 ± 0.069[b]** | **75.1** |
| Cefotaxime | 0 | 26.0 ± 1.8 | 3.44 ± 0.12 | 7.55 | 50.7 ± 6.6 | 6.57 ± 0.96 | 7.72 | 22.9 ± 0.5 | 0.657 ± 0.084[b] | 34.9 |
| | 50 | **16.7 ± 1.5** | **1.99 ± 0.09** | **8.39** | **58.9 ± 5.4** | **4.54 ± 0.74** | **13.0** | **15.1 ± 1.2** | **0.646 ± 0132[b]** | **23.4** |
| Ceftazidime | 0 | 14.2 ± 0.8 | 38.1 ± 1.6 | 0.373 | 2.02 ± 0.30 | 27.5 ± 3.4 | 0.0732 | 17.8 ± 0.3 | 19.7 ± 0.80 | 0.907 |
| | 50 | **14.7 ± 1.5** | **54.7 ± 5.5** | **0.269** | **1.40 ± 0.14** | **12.5 ± 1.7** | **0.112** | **11.7 ± 1.7** | **17.0 ± 1.9** | **0.691** |
| Cephalexin | 0 | 75.1 ± 8.8 | 18.5 ± 2.9 | 4.06 | 1.19 ± 0.08 | 5.47 ± 0.53 | 0.217 | 162 ± 19 | 10.4 ± 2.2 | 15.6 |
| | 50 | **104 ± 11** | **13.9 ± 1.7** | **7.52** | **1.41 ± 0.09** | **6.30 ± 0.72** | **0.224** | **160 ± 17** | **6.57 ± 0.81** | **24.4** |
| Meropenem | 0 | 25.1 ± 4.4 | 83.3 ± 16.1 | 0.301 | 6.79 ± 0.80 | 4.21 ± 0.29 | 1.63 | 3.65 ± 0.21 | 46.9 ± 2.2 | 0.0778 |
| | 50 | **26.2 ± 2.1** | **40.4 ± 4.6** | **0.649** | **27.3 ± 1.9** | **6.38 ± 0.63** | **4.28** | **2.81 ± 0.22** | **72.1 ± 7.7** | **0.0390** |
| Imipenem | 0 | 87.6 ± 4.5 | 48.6 ± 10.2 | 1.88 | ND | ND | 0.269 | 105 ± 6 | 116 ± 8 | 0.902 |
| | 50 | **101 ± 14** | **23.5 ± 3.2** | **4.31** | **ND** | **ND** | **0.529** | **127 ± 3** | **77.6 ± 1.9** | **1.64** |

[a]Values obtained in the presence of 50 µM ZnCl$_2$ are shown in bold. ND: not determined (the $K_m$ was too large).
[b]The $K_m$ value was determined using nitrocefin as the reporter substrate.

Among the 42 amino acid residues different between IMP-27 and IMP-1, most substitutions occurred on the surface of the molecule; seven residues, those at positions 58, 62, 68, 198, 226, 261, and 262, were close to the active site. We then prepared IMP-27 mutants by replacing each of these seven amino acids with the corresponding amino acids in IMP-1 and performed kinetic assays to investigate their roles (Table 1; Table S3). Among them, the one that showed the most significant change in kinetic properties was the G262S mutant, which was catalytically more efficient than IMP-27 against most of the investigated substrates, with kinetic properties comparable to those of IMP-1. In the presence or absence of Zn(II), the G262S mutation exhibited an increased $k_{cat}$ value for benzylpenicillin by four- to sevenfold and reduced $K_m$ value by less than a quarter of that of IMP-27. These changes significantly increased the enzyme's catalytic efficiency, comparable to that of IMP-1. The catalytic efficiency of the IMP-27 G262S mutant for cephalexin increased by two orders of magnitude due to an increase in the $k_{cat}$. In contrast, the G262S mutant showed lower catalytic efficiency for meropenem than IMP-27 and even IMP-1 due to its low $k_{cat}$ and high $K_m$ values.

## Susceptibility profiles

Minimum inhibitory concentration (MIC) measurement was performed using *E. coli* DH5α cells harboring the plasmids that have IMP-encoding genes in pMW119_Kan$^R$_Ptac, a low-copy cloning vector constructed from pMW119. It showed that cells expressing IMP-27 were more resistant to meropenem than cells expressing its G262S mutant (Table 2). In the presence of EDTA, the cells expressing IMP-27 were still more resistant to meropenem, while the G262S mutant gives higher resistance to cephalothin and ceftazidime.

## Effect of Zn(II) on enzyme stability

Thermal stability analysis *via* differential scanning fluorimetry (DSF) in the absence of Zn(II) revealed that IMP-27 and its G262S mutant had $T_m$ values of 68.5°C and 66.5°C, respectively, which were lower than those of IMP-1 (70.5°C) (Fig. S2). The $T_m$ values of IMP-27 and the G262S mutant decreased by 1.5°C in the presence of 50 µM Zn(II), while that of IMP-1 increased by 1.8°C with the addition of 50 µM Zn(II).

**TABLE 2** Antibiotic susceptibility of the IMP-type enzymes[a]

| | MIC (µg/mL) | | | | | |
|---|---|---|---|---|---|---|
| | Ampicillin | Benzylpenicillin | Cephalothin | Cefotaxime | Ceftazidime | Meropenem |
| DH5a | 4 (4) | 32 (32) | 4 (4) | 0.5 (0.25) | 16 (8) | ≤0.125 (≤0.125) |
| pMW119_KanR_Ptac | 4 (4) | 64 (32) | 8 (8) | 1 (1) | 16 (16) | ≤0.125 (≤0.125) |
| pMW119_KanR_Ptac_IMP27+ss | 64 (4) | 128 (32) | >128 (16) | >128 (4) | >128 (2) | 8 (1) |
| pMW119_KanR_Ptac-IMP27+ss G262S | >128 (8) | >128 (32) | >128 (64) | >128 (4) | >128 (8) | 1 (≤0.125) |
| ATCC25922 | 8 (8) | 64 (64) | 16 (16) | ≤0.125 (≤0.125) | 0.5 (0.5) | ≤0.125 (≤0.125) |
| ATCC27853 | >128 (>128) | >128 (>128) | >128 (>128) | 16 (16) | 2 (4) | 0.5 (1) |

[a]Values measured in the presence of 50 µM EDTA are shown in parentheses.

## Effect of chelating agents on enzymatic activity

The inhibitory effects of four chelating agents, EDTA, 4-(2-pyridyl azo) resorcinol (PAR), DPA, and L-captopril were analyzed by measuring the residual enzyme activity after the addition of these agents (Fig. 2). IMP-27 showed higher tolerance to EDTA and PAR than IMP-1 (Fig. 2A and B). PAR inhibited IMP-1 activity by up to 48% at a concentration of 0.5 mM, whereas it had no inhibitory effect on IMP-27, even at a concentration of 2 mM. DPA tended to inhibit IMP-1 more strongly than IMP-27 at low concentrations, although it inhibited IMP-27 more efficiently than IMP-1 at concentrations >0.2 mM (Fig. 2C). Introducing the G262S mutation in IMP-27 significantly changed the inhibition profiles of EDTA and PAR to levels comparable to those of IMP-1.

The number of Zn(II) ions per molecule was determined using inductively coupled plasma atomic emission spectroscopy (ICP-AES) (Fig. 3). Without chelating agents, the Zn(II) contents of IMP-1, IMP-27, and IMP-27G262S were 2.1 ± 0.2, 2.4 ± 0.2, and 2.0 ± 0.2, respectively. Treatment with EDTA resulted in a concentration-dependent decrease in the Zn(II) content. No significant difference was observed between IMP-1 and IMP-27, whereas IMP-27G262S readily lost Zn(II) ions due to precipitation from EDTA treatment at a concentration >0.2 mM (Fig. 3A). More Zn(II) ions remained bound to IMP-27 than to IMP-1 after DPA treatment (Fig. 3B).

## DISCUSSION

MBLs evolve in environments containing newly introduced antibiotics. IMP-27 is an IMP-type enzyme that was clinically isolated 20 years after the isolation of IMP-1, the first identified IMP-type enzyme. IMP-27 has been reported to have a lower catalytic efficiency than IMP-1 but a higher resistance to EDTA (19). In this study, we conducted structural determination analysis, kinetic analysis in the presence or absence of excess Zn(II) ions, and chelating agent resistance analysis of IMP-27, its mutant G262S, and IMP-1.

Crystallographic analysis revealed no significant difference in the overall structure between IMP-27 and other IMP-type enzymes. However, looking at the active sites of these enzymes, the positions of Zn2 varied more between IMP-27 and most other IMP-type enzymes than that of Zn1. We observed that the position of Zn2 in IMP-27 was similar to that of IMP-6, which had a glycine residue at position 262, the same as IMP-27. The glycine residue at position 262 could affect the position of Zn2 by providing flexibility to the adjacent His263 residue.

In this study, excess Zn(II) significantly increased the catalytic turnover rate of IMP-27 toward meropenem and slightly increased the affinity toward several cephalosporins. However, similar changes were not observed in IMP-1, although it did cause a slight increase in $K_m$ and a decrease in $k_{cat}/K_m$ for penicillins and cephalosporins (Table 1). The decreased activity of IMP-27 in the absence of excess Zn(II) may be due to the dissociation of the metal ion from the active site, as reported for the C221D mutant of the BcII enzyme (21). Considering the increase in the $T_m$ of IMP-1 in the presence of 50 µM Zn(II), additional Zn ions may bind to the enzyme to stabilize its conformation and make the substrate-binding site less flexible. Zn ions bound to IMP-1 rapidly exchange

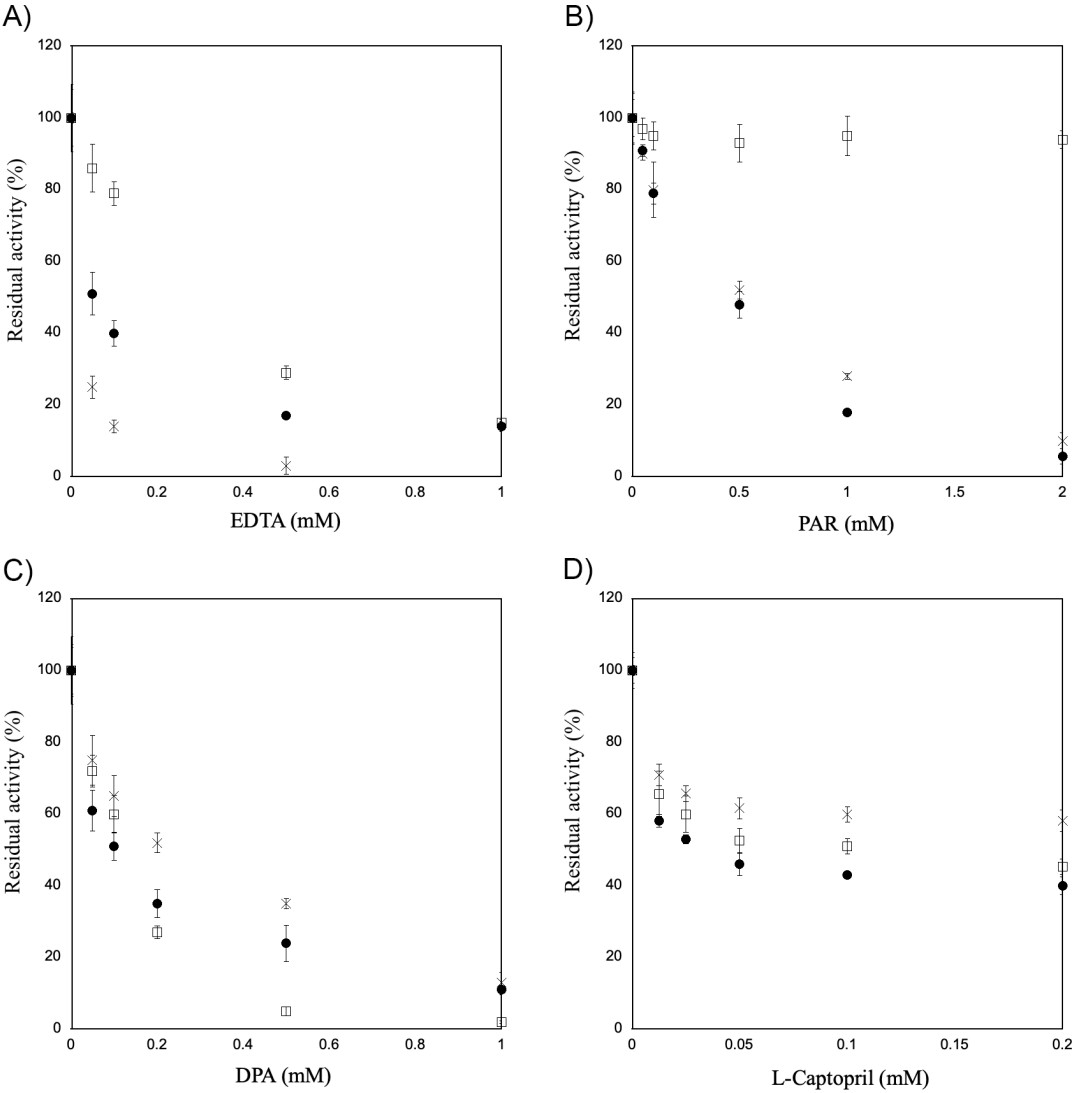

**FIG 2** Effect of chelating agents on enzyme activity. Residual enzyme activity in the presence of (A) EDTA, (B) 4-(2-pyridyl azo) resorcinol (PAR), (C) 2,6-dipicolinic acid (DPA), and (D) L-captopril. The residual activities of IMP-1 (●), IMP-27 (□), and IMP-27G262S (×) are shown.

with extraneous $Zn^{2+}$, suggesting that excess $Zn^{2+}$ inhibits the catalytic activity of IMP-1 by binding in the vicinity of the active site (22).

The residue at position 262 is key to determining the substrate specificity of IMP-type enzymes. Oelschlaeger et al. studied IMP-1 and IMP-6 (the S262G mutant of IMP-1). They showed that the G262S mutation resulted in equal or reduced activity toward "type I substrates," such as cephalothin, cefotaxime, meropenem, and doripenem, and improved catalytic efficiency toward "type II substrates," including cephaloridine, ceftazidime, penicillins, and imipenem (23, 24). The results of our kinetic study on IMP-27 were generally consistent with their results. The opposite effect of the G262S mutation on the hydrolysis of two carbapenem antibiotics, imipenem and meropenem, should be related to the fact that imipenem is a type II substrate with a positively charged R2 group, whereas meropenem is a type I substrate with a neutral R2 group of a high electron density. The catalytic efficiency of IMP-27 toward meropenem was approximately sixfold higher than that of IMP-1, whereas the catalytic efficiency of IMP-1 and IMP-6 toward meropenem was comparable (24). This implies that residues other than those different between IMP-1 and IMP-27 also affect the kinetic properties of IMP-27.

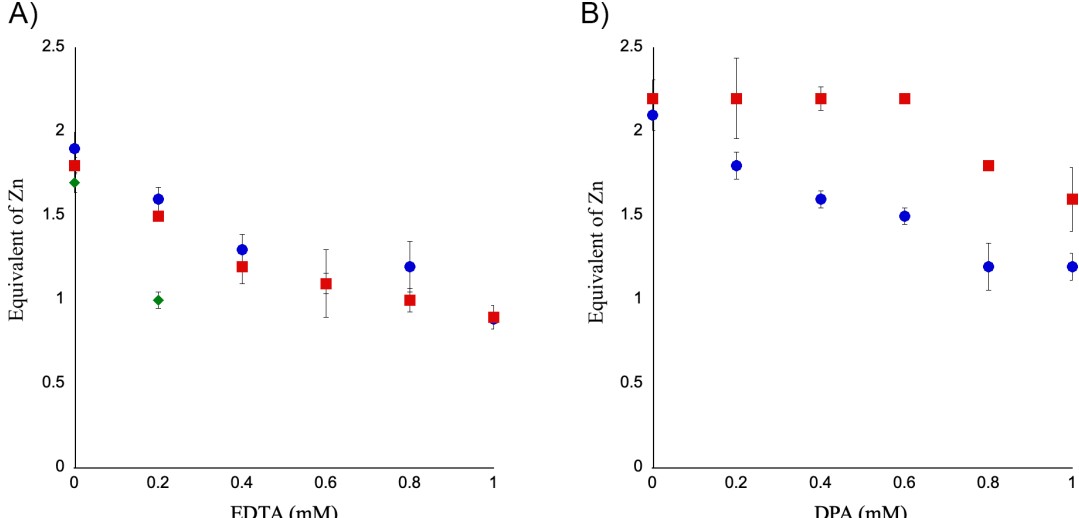

**FIG 3** Change in zinc (Zn) content upon the addition of chelating agents. The number of Zn(II) ions bound to the enzyme was determined using ICP-AES. The enzymes were treated with 0–1,000 µM (A) EDTA and (B) DPA. The Zn(II) content of the IMP-27 G262S mutant was determined only with <0.2 mM EDTA because the enzyme precipitated upon treatment with EDTA at concentrations >0.2 mM. The horizontal axis indicates the concentration of each chelating agent, while the vertical axis indicates the number of Zn(II) per enzyme molecule. ●-IMP-1, ■-IMP-27, ◆-IMP-27G262S.

The *in vitro* assay showed that IMP-27 had a higher tolerance to EDTA than IMP-1 based on the analysis of the residual enzyme activity after the addition of the chelating agent (Fig. 2), while ICP-AES analysis showed that the number of Zn(II) ions per molecule after EDTA treatment was not significantly different between IMP-27 and IMP-1 (Fig. 3). Between the Zn ions Zn1 and Zn2, the one that is more critical for enzyme activity may remain bound in IMP-27 rather than in IMP-1.

Kinetic analysis of the IMP-27 G262S mutant indicated that residue 262 was key in determining the properties of IMP-27. The G262S mutant was more catalytically efficient than IMP-27 for most substrates. This result is generally consistent with that of a previous study on IMP-6, which has a glycine residue at position 262 as IMP-27. Oelschlaeger et al. proposed that this glycine residue increases the flexibility of the adjacent His263 residue, causing a decrease in enzymatic activity against benzylpenicillin, ampicillin, and ceftazidime (23, 24). The residue at position 262 significantly affects the position of Zn2 in the crystal structures of IMP-type enzymes. Specifically, the G262S mutation drastically reduced the Zn(II)-binding capacity of IMP-27 (Fig. 1B and 2A). These results indicate that this residue has a significant effect on the orientation and/or flexibility of His263, which binds to Zn2. The G262S mutation altered the kinetic properties of IMP-27 toward penicillins and imipenem to be comparable to IMP-1. However, this mutant was catalytically more efficient than IMP-1 for four substrates tested in this study (cephalothin, cefotaxime, ceftazidime, and cephalexin), which had no obvious similarities in their R2 groups. This result indicates that other residues of IMP-27 that differ from IMP-1, together with Ser262, increase the catalytic efficiency for these substrates.

The most striking features of IMP-27 are its high hydrolytic activity toward meropenem, even in the presence of 50 µM Zn(II). Meropenem is a β-lactam agent with one of the largest R2 groups. The crystal structure of IMP-13 in complex with meropenem hydrolysate (PDB ID: 6RZS) indicates that the L3 and L10 loops and residues Val61, Trp64, Val67, Asp120, His196, Lys224, Gly227, and Asn230 are important for binding to meropenem (25). Because the residues listed above are well-conserved in IMP-type enzymes, including IMP-27 and IMP-1, the high hydrolytic activity of IMP-27 toward meropenem could be caused by other factors.

By analyzing the crystal structure of IMP-27, we found a hydrogen bond between the side chains of Ser261 and Ser264 in the L12 loop (residues 262–268) (Fig. 4A). This hydrogen bond was not observed in the crystal structures of other IMP-type enzymes.

IMP-1 and IMP-6 contained a Pro at position 261 that does not form a hydrogen bond (Fig. S1; Fig. 4B). Ser261 and Ser264 were conserved in IMP-2, IMP-13, and IMP-18; however, no hydrogen bonds formed between the side chains of these two residues (Fig. 4C). Comparisons of the primary sequences and structures implied that the formation of this hydrogen bond may be affected by the residue at position 262, which is also in the L12 loop. IMP-2, IMP-13, and IMP-18 contained serine at position 262, the same as IMP-1 (Fig. S1). Therefore, a hydrogen bond between Ser261 and Ser264 may only be formed if position 262 is occupied by a glycine residue. Preliminary kinetic analysis showed that the S261P mutation did not cause a drastic change in kinetic parameters toward meropenem, suggesting that the formation of this hydrogen bond is not the only critical factor for the meropenem recognition and hydrolysis (Table S3).

We also analyzed the inhibitory effect of four chelating agents by measuring the residual enzyme activity after adding these agents. EDTA, DPA, and PAR are inhibitors that work through metal ion stripping, while captopril is an inhibitor that functions through the formation of the ternary complex MBL: zinc: inhibitor (26–29). IMP-27 had a higher tolerance to EDTA than did IMP-1 (Fig. 2A), suggesting that the metal ion(s) essential for enzymatic activity is/are more easily stripped by chelating agents in IMP-1 than in IMP-27. However, a significant difference in the zinc content was not observed between these enzymes after treatment with EDTA (Fig. 3A). The IMP-27 G262S mutant was remarkably more susceptible to EDTA than IMP-1 and wild-type IMP-27, consistent with the results of the metal content analysis. This analysis revealed that the G262S mutant readily lost Zn(II) ions and precipitated following EDTA treatment at a concentration >0.2 mM (Fig. 3A). PAR showed little inhibitory effect on IMP-27, even at a concentration of 2 mM; however, it inhibited the G262S mutant as well as IMP-1 (Fig. 2B). This suggests that the existence of glycine at position 262 reduces the accessibility of PAR to the zinc ions, which may be related to subtle changes in the position of Zn2 and its ligand residue His263 observed in the crystal structure, resulting in high tolerance toward PAR. DPA inhibited IMP-27 more efficiently than IMP-1 at concentrations > 0.2 mM (Fig. 2C), implying that DPA is a strong inhibitor that can sufficiently remove metal ions from IMP-27. The result with L-captopril suggests that it forms a ternary complex with all three enzymes (Fig. 2D). Subtle differences between the enzymes suggest that the formation of this complex is slightly easier with IMP-1 and slightly harder or more unstable with IMP-27 and its G262S mutant.

A mutation at position 262 was reported to affect the properties of other MBL enzymes aside from IMPs. Tomatis et al. showed that the G262S mutation in an MBL from *Bacillus cereus* (BcII) enhanced enzymatic activity through the better positioning of Zn2 and subsequent stabilization of a negatively charged reaction intermediate (30). Nuclear magnetic resonance analysis of BcII indicated that the G262S mutation acted as a switch to alter the slow timescale conformational dynamics in the loops surrounding the active site, enabling a larger substrate spectrum (31). The results of our kinetic analysis is generally consistent with those of BcII and its mutants, indicating the possibility that the G262S mutation in IMP-type enzymes causes similar effects on the conformation of the enzyme near the active site. Crystal structure comparison between the wild-type BcII and the M5 variant that contains four mutations (N70S, V112A, L250S, and G262S) showed differences in the conformation of the loops surrounding the active site and a significant positional shift of the Zn2 closer to Zn1 in the mutant; neither of these was observed between IMP-27 and IMP-1. This may be due to the absence of the N70S mutation that gives flexibility to the loops in the M5 mutant of BcII or the difference in the loop regions between IMP-type enzymes and BcII (Fig. 1B; Table S2) (11). Considering these comparisons, it is speculated that the effect of the G262S mutation may not be the same between IMPs and BcII.

In this study, we compared IMP-27 with IMP-1 and found significant differences in their substrate specificity, tolerance to chelating agents, and metal binding. Our results indicate that IMP-27 evolved to obtain high affinity and catalytic efficiency against meropenem at the expense of catalytic efficiency against penicillins. In addition, IMP-27

might have improved metal-binding capacity, maintaining its activity under Zn(II)-deficient conditions better than IMP-1. Our results revealed that residue 262 is one of the key determinants of the enzymatic properties of IMP-27. Moreover, they suggest that other differential residue(s) between IMP-27 and IMP-1, possibly in combination with residue 262, can alter the kinetic properties of these enzymes for certain substrates.

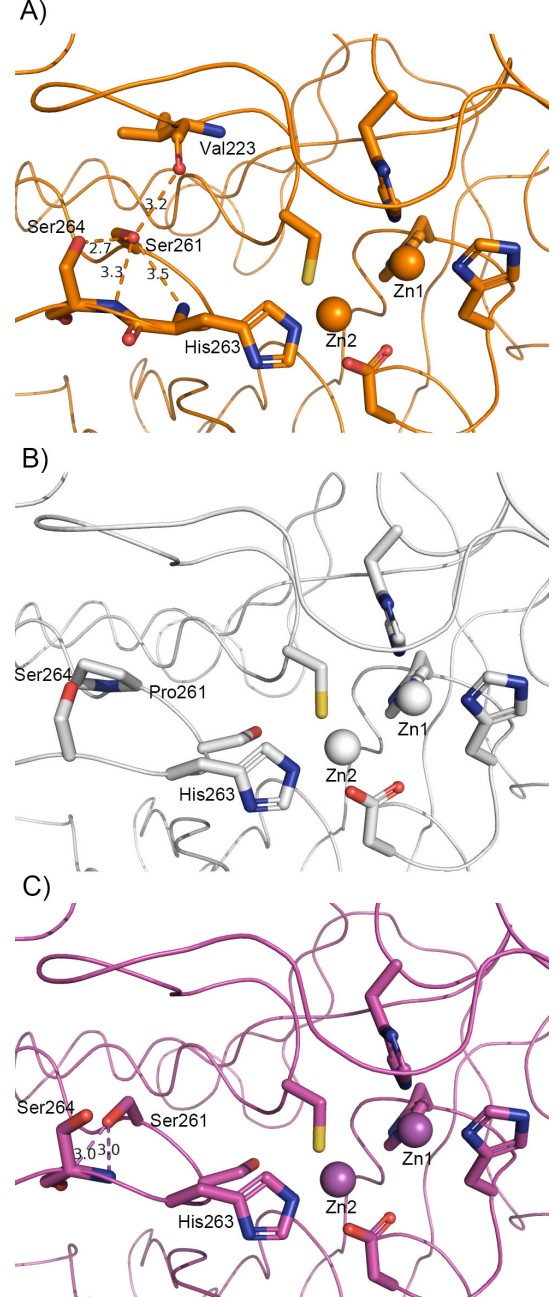

**FIG 4** Comparison of hydrogen bonds involving the side chain of amino acid 261 in IMP-type metallo-β-lactamases. (A) The structure of IMP-27 (PDB ID: 6L3S) is shown in orange. (B) The structure of IMP-1 (PDB ID: 5Y5B) is shown in white. (C) The structure of IMP-13 (PDB ID: 6R79) is shown in magenta. The Zn(II) ions in each enzyme are shown in their respective colors. Hydrogen bonds are indicated by dotted lines with their respective distances (Å).

## MATERIALS AND METHODS

### Plasmid construction

For the MIC analysis, we used pMW119_KanR_Ptac, a low-copy cloning vector constructed from pMW119 (GenBank accession no: AB005476) by replacing its ampicillin resistant gene with kanamycin resistant gene, and by replacing its *lac* promotor and *lac*Z encoding regions with tac promoter. The genes encoding the IMP enzymes with or without signal sequences were cloned into this vector. The signal sequence of IMP-27 was replaced with that of IMP-1. An expression plasmid of mature IMP-27 with the signal sequence of IMP-1 was constructed and designated pMW119_KanR_Ptac_IMP27+ss.

The genes encoding the mature sequences of IMP-27 and IMP-1 were inserted into the vector pET-26b to obtain pET-26b-IMP-27 and pET-26b-IMP-1, respectively, which express the respective enzymes containing the *pelB* signal sequence. The enzymes expressed by these plasmids were used for enzymatic assays and metal content analysis.

For the crystallographic analysis, the vector pET-28a-IMP-27Cth was used to express IMP-27, which contains a $(His)_6$ tag at the C-terminus but without a signal sequence. Plasmids that expressed the IMP-27 G262S mutant were constructed *via* site-directed mutagenesis using a PrimeSTAR Mutagenesis Basal kit (TaKaRa Bio Co., Shiga, Japan). Two primers, IMP-27-G262S-for (5′-GTTGTCTCGAGTC ATAGCGAAACCGGAG-3′) and IMP-27-G262S-rev (5′-ATGACTCGAGACAACCAGCTTTGCTT TA-3′) were used to introduce the mutation to the plasmids pET-26b-IMP-27 and pMW119_KanR_Ptac_IMP27+ss, generating the plasmids pET-26b-IMP-27G262S and pMW119_KanR_Ptac_IMP27+ssG262S, respectively.

### Expression and purification of IMP-27 for crystallization

To crystallize IMP-27, *E. coli* BL21(DE3) pLysS (Promega, Madison, WI, USA) cells harboring the pET28a-IMP-27Cth plasmid were cultured in 100 mL 2 × TY medium at 37°C. Protein production was induced with 0.5 mM isopropyl β-D-1-thiogalactopyranoside when the $OD_{600}$ of the culture reached approximately 0.5. After further incubation at 18°C for 12 h, the cells were harvested *via* centrifugation, suspended in 50 mM sodium phosphate buffer (pH 7.0) containing 500 mM NaCl, and disrupted *via* sonication. The supernatant was loaded onto a 1 mL HisTrap FF column (Cytiva, Marlborough, MA, USA) equilibrated with the same buffer, and the protein was eluted with imidazole at a 0–500 mM linear gradient. The sample buffer was changed to 20 mM HEPES buffer (pH 7.5) containing 50 µM $ZnCl_2$ using a PD-10 column (Cytiva). Subsequently, 3 U of thrombin (Cytiva) was added to every 1 mg of the purified protein and incubated at 10°C for 24 h to cleave the $(His)_6$ tag. Thrombin was removed using a HiTrap Benzamidine FF column (Cytiva) equilibrated with 50 mM Tris-HCl buffer (pH 8.0) containing 500 mM NaCl. The fraction containing IMP-27 was then loaded onto a HisTrap FF column equilibrated with 50 mM sodium phosphate buffer (pH 7.0) containing 500 mM NaCl to exclude the cleaved $(His)_6$-tag and remaining $(His)_6$-tagged proteins. The sample was then loaded onto a PD-10 column equilibrated with 20 mM HEPES buffer (pH 7.5) for buffer exchange and concentrated using an Amicon Ultra 3K centrifugal filter unit to a final concentration of 10 mg mL$^{-1}$.

### Enzyme expression and purification for activity measurements and metal content analysis

We purified the enzymes secreted into the periplasm for use in the enzymatic measurements and metal content determination. IMP-1 and IMP-27 were produced in *E. coli* BL21(DE3) cells harboring the pET-26b-IMP-27 and pET-26b-IMP-1 plasmids, respectively. The cells were cultured in 100 mL 2 × TY medium at 37°C. Protein expression was induced with 0.1 mM isopropyl β-D-1-thiogalactopyranoside when the $OD_{600}$ of the culture reached approximately 0.5, and the cultures were further incubated at 30°C for 20 h. The cells, which were harvested *via* centrifugation, were suspended in 30 mL of

50 mM Tris-HCl (pH 8.0) containing 20% sucrose and incubated at 25℃ for 10 min. The cells were collected *via* centrifugation for 10 min at 10,000 × *g* at 4℃, resuspended in 30 mL of 5 mM $MgSO_4$, and incubated on ice for 10 min. The cells were then centrifuged for 10 min at 10,000 × *g* at 4℃ to collect the supernatant as the periplasmic fraction, which was subjected to purification. IMP-27 was purified using TOYOPEARL SuperQ-650M resin (TOSOH Co., Japan) equilibrated with 50 mM glycine-NaOH buffer (pH 9.0) and eluted with a buffer containing 50 mM NaCl. IMP-1 was purified using a HiTrap SP FF column (Cytiva) equilibrated with 50 mM Tris-HCl (pH 7.0) and eluted with NaCl at a 0–500 mM linear gradient. These purified enzymes were applied to a PD-10 column equilibrated with 20 mM HEPES buffer (pH 7.5) for buffer exchange and concentrated with an Amicon Ultra-15 (Merck) to a final concentration of 1 mg mL$^{-1}$. Enzyme purity was verified *via* sodium dodecyl sulfate-polyacrylamide gel electrophoresis using 15% acrylamide gels.

## Crystallization, data collection, and structure determination

The crystallization conditions were initially screened using the sitting-drop vapor-diffusion method with a Crystal Screen and Crystal Screen II (Hampton Research, CA, USA). After optimizing the crystallization conditions, crystals suitable for data collection were grown using the hanging-drop vapor-diffusion method as follows: 5 µL of 10 mg mL$^{-1}$ IMP-27 was mixed with an equal volume of a reservoir solution containing 0.3 M $MgCl_2$, 0.1 M Tris-HCl (pH 9.0), and 30% (w/v) polyethylene glycol-4000, followed by incubation at 10℃ for 1 week. The obtained crystals were cryoprotected by adding 18% ethylene glycol to the reservoir solution and subjected to X-ray diffraction using the beamline BL-5A of the Photon Factory, KEK (Tsukuba, Japan). The diffraction patterns were indexed, integrated, and scaled using iMosflm (32) and SCALA from the CCP4 suite (33). The search model for molecular replacement was generated using SWISS-MODEL (34) based on the primary sequence of IMP-27 and the crystal structure of IMP-2 (PDB ID: 4UBQ). Molecular replacement was performed using MOLREP (35), and the model was further refined using COOT and Refmac (36, 37). The protein interface was analyzed using Protein Interfaces, Surfaces, and Assemblies (PISA, https://www.ebi.ac.uk/pdbe/pisa/) (38).

## Enzyme activity measurements

Steady-state kinetic analysis was performed using the following substrates: ampicillin, cephalothin, ceftazidime (Sigma-Aldrich, USA); benzylpenicillin and cefotaxime (FUJIFILM Wako Chemicals, Japan); cephalexin, meropenem, and aztreonam (Tokyo Chemical Industry); and imipenem (TargetMolChemicals, Boston, MA, USA) (Table S4). The enzyme reaction was performed at 30℃ in 500 µL of 20 mM HEPES buffer (pH 7.5) containing 0–50 µM $ZnCl_2$ and detected using a UV-1900i UV-VIS Spectrophotometer (Shimadzu, Japan). The same buffer containing 50 µg mL$^{-1}$ bovine serum albumin and 0.01% Triton X-100 was used as the enzyme dilution buffer. Kinetic analysis was performed as described by Borgianni *et al.* (39). $K_m$ values lower than 1 µM were determined using 50 µM nitrocefin as the reporter substrate. All enzymatic measurements were performed at least three times.

## Susceptibility profiles

*E. coli* DH5α cells harboring the plasmids that have IMP-encoding genes in pMW119_Kan$^R$_Ptac, a low-copy cloning vector constructed from pMW119, were cultured in cation-adjusted Mueller–Hinton broth (Becton, Dickinson and Company, NJ, USA). Antibiotic susceptibility testing was performed using the microbroth dilution method according to the guidelines of the Clinical and Laboratory Standards Institute (40). The MICs were also determined in the presence of 50 µM EDTA (Fujifilm Wako Chemicals) to simulate conditions with limited Zn(II) availability.

## Equilibrium dialysis and metal content determination using ICP-AES

Equilibrium dialysis was performed as described by Chen *et al.* (41). The chelating agents used in this study were EDTA, DPA (Fujifilm Wako Chemicals), L-captopril (Fujifilm Wako Chemicals), and 4-(2-pyridyl azo) resorcinol (PAR) (Sigma-Aldrich). Briefly, 500 µL of a 119.4 µM enzyme solution was diluted 1:2 (v/v) with 20 mM HEPES buffer (pH 7.5) containing 0–2.0 mM of the chelating agent. After incubation for 30 min at 25℃, the solutions were dialyzed versus 500 mL of metal-free 20 mM HEPES buffer (pH 7.5) for 12 h at 4℃ using a Spectra/Por 3 Dialysis Tubing 3.5 kD MWCO (Repligen, USA). The metal content of the samples was analyzed *via* ICP-AES using an ICPE-9000 system (Shimadzu). The calibration curve was prepared using an ICP multi-element standard solution IV (Merck). The concentration of Zn(II) was quantified at 202.548 nm. An yttrium standard solution (Merck) was added to the samples at a final concentration of 1.0 µg mL$^{-1}$ as an internal standard. All samples and standard solutions were prepared in 1.0 N HNO$_3$.

## DSF measurements

The DSF measurement was performed using a CFX96 Touch Real-Time PCR Detection System (Bio-Rad). SYPRO-Orange (Invitrogen, USA) at a final concentration of 5× was added into 12.5 µL of the reaction buffer, which contained 20 mM HEPES buffer (pH 7.5), 0–50 µM ZnCl$_2$, and 0.4 mg/mL of the purified enzymes. Analysis was carried out using the FRET scan mode at temperatures ranging from 10℃ to 95℃, increasing the temperature by 1℃ min$^{-1}$. Data were analyzed using GraphPad Prism ver 5.01 software (GraphPad Software, USA).

## Enzyme inactivation using chelating agents

The resistance of IMP-type enzymes to chelating agents was analyzed as described by Valladares *et al.* (42). Briefly, 0.1 mg mL$^{-1}$ of each enzyme diluted with the enzyme dilution buffer containing 0–2 mM chelator was incubated at 25℃ for 30 min. Enzyme activity was then measured using 80 µM cefotaxime as a substrate.

### ACKNOWLEDGMENTS

We thank Yu-ta Maeda and Kentaro Hoshi for their assistance with the preparation of the mutant enzymes and MIC measurements and Takateru Yamada for sample preparation for metal content analysis using ICP-AES.

This work was supported by JSPS KEKENHI Grant Number JP19K07566 (to A.S.-I.) and an Examination Research Fund Grant from The Uchida Energy Science Promotion Foundation (to A.S.-I.).

### AUTHOR AFFILIATIONS

[1]Department of Applied Life Sciences, Niigata University of Pharmacy and Applied Life Sciences, Niigata, Japan
[2]Data4cs Kabushiki Kaisha (Data4cs K.K.), Tokyo, Japan
[3]Graduate School of Science, Kanagawa University, Yokohama, Japan
[4]Faculty of Pharmacy, Niigata University of Pharmacy and Medical and Life Sciences, Niigata, Japan
[5]Department of Microbiology and Infectious Disease, Toho University School of Medicine, Tokyo, Japan
[6]Microbial Genomics and Ecology, The Center for Planetary Health and Innovation Science, The IDEC Institute, Hiroshima University, Hiroshima, Japan

### AUTHOR ORCIDs

Yoshiki Kato  http://orcid.org/0000-0003-0105-6930
Toshio Yamaguchi  http://orcid.org/0009-0008-3256-1538

Yoshikazu Ishii http://orcid.org/0000-0002-1943-4648

Akiko Shimizu-Ibuka http://orcid.org/0000-0002-5310-9216

## FUNDING

| Funder | Grant(s) | Author(s) |
|---|---|---|
| MEXT \| Japan Society for the Promotion of Science (JSPS) | JP19K07566 | Akiko Shimizu-Ibuka |
| Uchida Energy Science Promotion Foundation (内田科学振興財団) | | Akiko Shimizu-Ibuka |

## AUTHOR CONTRIBUTIONS

Yoshiki Kato, Data curation, Investigation, Writing – original draft | Toshio Yamaguchi, Investigation, Methodology, Validation, Writing – review and editing | Haruka Nakagawa-Kamura, Investigation, Methodology, Supervision, Writing – review and editing | Yoshikazu Ishii, Conceptualization, Funding acquisition, Methodology, Project administration, Supervision, Writing – review and editing | Akiko Shimizu-Ibuka, Conceptualization, Funding acquisition, Investigation, Methodology, Project administration, Supervision, Writing – review and editing

## DATA AVAILABILITY

The authors confirm that the data supporting the findings of this study are available within the article and its supplementary materials. The structure and the corresponding structure factor amplitudes of IMP-27 are available at the RCSB PDB under accession code 6L3S.

## ADDITIONAL FILES

The following material is available online.

### Supplemental Material

**Supplemental material (Spectrum00391-24-s0001.docx).** Tables S1 to S4, and Figures S1 and S2.

### Open Peer Review

**PEER REVIEW HISTORY (review-history.pdf).** An accounting of the reviewer comments and feedback.

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
