## [Reviewer comments · Microbiology Spectrum]

Microbiology Spectrum

Functional and structural analysis of IMP-27 metallo- β -lactamase: Evolution of IMP-type enzymes to overcome Zn(II) deprivation

Yoshiki Kato, Toshio Yamaguchi, Haruka Nakagawa-Kamura, Yoshikazu Ishii, and Akiko Shimizu-Ibuka

Corresponding Author(s): Akiko Shimizu-Ibuka, Kanagawa Daigaku

Review Timeline:

Submission Date:	February 20, 2024
Editorial Decision:	March 8, 2024
Revision Received:	October 7, 2024
Accepted:	October 9, 2024

Editor: Pablo Power

Reviewer(s): The reviewers have opted to remain anonymous.

Transaction Report:

DOI: <https://doi.org/10.1128/spectrum.00391-24>

Re: Spectrum00391-24 (Functional and structural analysis of IMP-27 metallo- β -lactamase: Evolution of IMP-type enzymes to overcome Zn(II) deprivation)

Dear Dr. Akiko Shimizu-Ibuka:

Thank you for the privilege of reviewing your work. Below you will find my comments, instructions from the Spectrum editorial office, and the reviewer comments.

Revision Guidelines

Sincerely,
Pablo Power
Editor
Microbiology Spectrum

Reviewer #1 (Comments for the Author):

The authors have considered and addressed most of the comments and criticisms raised by the reviewers rather satisfactorily. There are three aspects that they also may want to consider:

1. Regarding comment 7, I find the authors' answer unsatisfactory. If they have indeed performed kinetic measurements on the mutants and they have retrieved kinetic constants with properly evaluated errors, they must be indicated in the text rather than

mentioned as "data not shown" as they propose. It is not possible that the values are identical, so they should show them. If the authors, instead, have done preliminary kinetic measurements that suggest that the mutants have similar kinetic behavior, and they have not retrieved, kinetic constants, they should indicate this.

2. Regarding the G262S mutation, the authors may also want to discuss their results at the light of the impact of a similar mutation in the B1 enzyme BclI, which was analyzed in detail (doi: 10.1093/molbev/msw052, doi: 10.1073/pnas.0807989106)

3. Finally, they should work in polishing the language.

Reviewer #2 (Comments for the Author):

IMP-27 is a rare IMP enzyme that shares 82 % amino acid sequence identity with IMP-1. A previous report suggests low hydrolytic efficiency of imipenem and relative resistance to zinc chelators for IMP-27. Expanding upon previous findings, this study determined the structure, kinetics, and zinc dependence of IMP-27 and its mutant G262S in comparison with IMP-1. The overall structure was comparable to IMP-1 and other IMP-type enzymes. IMP-27 had lower catalytic efficiency against ampicillin, benzylpenicillin, and cephalexin but higher affinity and catalytic efficiency against meropenem than IMP-1 suggesting that the catalytic site of IMP-27 was optimized to hydrolyze meropenem in its molecular evolution at the expense of catalytic efficiency against penicillins. IMP-27 had a higher resistance to chelating agents than IMP-1. The authors attribute the differences in kinetic properties and chelator resistance between IMP-1 and IMP-27 to the G262S substitution.

The work nicely complements previous reports on G262S and IMP-27, and adds to the body of knowledge on this understudied but important group of acquired carbapenemase gene. The experimental details are well detailed and appear to be robust.

Some comments for the authors' consideration:

- The abstract states that the specificity of IMP-27 is tilted in favor of carbapenems, but the previous study by Dixon and colleagues reported lower specificity for imipenem. Please review and revise as appropriate. In addition, can structural insights be discussed regarding these differences between the two carbapenems?
- The importance section essentially repeats the abstract with findings. It should instead focus on the importance of the work and how it may contribute to better understanding of this unique class of enzymes.
- Line 33: Perhaps "residual" more appropriate than "remaining" here?
- Line 65: "MBLs have lower catalytic efficiency than SBLs" - can the authors provide reference(s)?
- Line 110: The term variable is somewhat ambiguous. Do the authors mean the conformation is different from IMP-1, or that the structure is flexible?
- Line 114: The term diverse is also unclear. Diverse means there is a variety within a group. Here the comparison is between IMP-27 and others, so one on one.
- Line 290: susceptible
- Line 318: Do we know the reason for the low expression?
- Line 320: I do not agree with the authors that it is appropriate to use an expression vector for MIC testing.
- Line 365: What was the centrifugation condition?

Thank you for all the useful comments and suggestions. We carefully considered all the comments from the reviewers and revised our manuscript accordingly. Considering the comments from Reviewer #2, additional MIC measurements were performed using a different plasmid vector. The manuscript was refined with the help of native English-speaking editors. Our responses to each of the comments are as follows:

Comments from Reviewer #1

Comment 1:

Regarding comment 7, I find the authors' answer unsatisfactory. If they have indeed performed kinetic measurements on the mutants and they have retrieved kinetic constants with properly evaluated errors, they must be indicated in the text rather than mentioned as "data not shown" as they propose. It is not possible that the values are identical, so they should show them. If the authors, instead, have done preliminary kinetic measurements that suggest that the mutants have similar kinetic behavior, and they have not retrieved, kinetic constants, they should indicate this.

Response:

Preliminary kinetic data for the mutants, which were measured against a limited number of substrates at defined concentrations, have been added as Table S3 in the Supplementary Material, and the text was amended accordingly (lines 139–140, page 10). The kinetic data for the S261P mutant were also included in Table S3; therefore, we amended the discussion on the function of the hydrogen bond between Ser261 and Ser264 accordingly (lines 267–270, page 17).

Comment 2:

Regarding the G262S mutation, the authors may also want to discuss their results at the light of the impact of a similar mutation in the B1 enzyme BcII, which was analyzed in detail (doi: 10.1093/molbev/msw052, doi: 10.1073/pnas.0807989106)

Response:

The text was revised to amend the corresponding section in the Discussion referring to the manuscripts suggested by the reviewer (lines 293–310, page 19).

Comment 3:

Finally, they should work in polishing the language.

Response:

The revised manuscript was checked and refined with the help of native English-speaking editors.

Reviewer #2 (Comments for the Author):

Comment 1:

The abstract states that the specificity of IMP-27 is tilted in favor of carbapenems, but the previous study by Dixon and colleagues reported lower specificity for imipenem. Please review and revise as appropriate. In addition, can structural insights be discussed regarding these differences between the two carbapenems?

Response:

Considering this comment, we amended the manuscript as follows:

- Kinetic data for imipenem were added to Table 1 in the revised manuscript. IMP-27 had lower catalytic efficiency for imipenem than IMP-1, as reported by Dixon *et al.* It was mentioned in the revised manuscript (line 30, page 3 and lines 125–126, page 9 in the revised manuscript).
- The abstract was amended to state that the specificity of IMP-27 changes in favor of meropenem only, not all carbapenems (line 33, page 3).
- We have also added descriptions of the structural difference between imipenem and meropenem (lines 217–221, pages 14–15).

Comment 2:

The importance section essentially repeats the abstract with findings. It should instead focus on the importance of the work and how it may contribute to better understanding of this unique class of enzymes.

Response:

We have revised this section according to the reviewer's recommendations (lines 40–50, page 4).

Comment 3:

Line 33: Perhaps "residual" more appropriate than "remaining" here?

Response:

This sentence was deleted in the revised manuscript, following the results of the MIC assay re-test, as described in the response to comment 9.

Comment 4:

Line 65: "MBLs have lower catalytic efficiency than SBLs" - can the authors provide reference(s)?

Response:

A reference was added in the revised manuscript (line 62, page 5).

Comment 5:

Line 110: The term variable is somewhat ambiguous. Do the authors mean the conformation is different from IMP-1, or that the structure is flexible?

Response:

In this sentence, we meant that the position of the His263 side chain was unique in IMP-27, compared to other Zn-ligand residues in IMP-type enzymes. We have amended this sentence in the revised manuscript (lines 105–107, page 8).

Comment 6:

Line 114: The term diverse is also unclear. Diverse means there is a variety within a group. Here the comparison is between IMP-27 and others, so one on one.

Response:

The word was replaced with "varied" in the revised manuscript (line 109, page 8).

Comment 7:

Line 290: susceptible

Response:

This word was amended in the revised manuscript according to the reviewer's suggestion (line 279, page 18).

Comment 8:

Line 318: Do we know the reason for the low expression?

Response:

In preliminary experiments, we noticed a very low expression level of IMP-27 that contains the native signal sequence; however, we did not investigate why that was the case. The signal sequences of IMP-1 and IMP-27 contained 18 residues, and 10 residues were identical. This difference could affect enzyme expression and signal peptide processing upon export to the periplasm in *E. coli*. No investigation into its causes has been carried out because it is beyond the scope of this study.

Comment 9:

Line 320: I do not agree with the authors that it is appropriate to use an expression vector for MIC testing.

Response:

We re-performed MIC measurements using *E. coli* DH5 α that expresses IMP-27 or its G262S mutant with a low-copy cloning vector pMW119_KanR_Ptac. The previous data indicated that IMP-27 had higher resistance to chelating agents than its G262S mutant and IMP-1, but the new results with this vector didn't show significant difference between the enzymes. We revised the manuscript as follows:

- The data in Table 2 were replaced with newly measured data.
- We deleted lines 31–34, page 2, and lines 225–229, page 14 in the previous manuscript, because the MIC measurement of *E. coli* expressing IMP-1 was not performed.
- We amended the manuscript (lines 151–157, page 11) due to the data replacement in Table 2.
- We deleted lines 251–252 (page 15) in the previous manuscript, because our new MIC data did not show significantly higher antibiotic resistance of the IMP-27 expressing cells than those expressing the G262S mutant under Zn(II)-depleted condition.
- We revised the description of plasmid construction in the Materials and Methods section (lines 323–330, page 21, and lines 342–343, page 22, in the revised manuscript).
- The measurements were performed according to the guidelines of the Clinical Laboratory Standards Institute, without modification of the method. We amended the Materials and Methods section (lines 426–428, pages 26-27).

- Dr. Haruka Nakagawa-Kamura, who performed the MIC measurements, was added to the authors.

Comment 10:

Line 365: What was the centrifugation condition?

Response:

We assumed that the reviewer pointed out the insufficient explanation for enzyme purification from the periplasmic fraction. We have added the centrifugation conditions and amended that paragraph (lines 378–381, page 23 in the revised manuscript).

Other points

Prof. Yoshikazu Ishii moved to a new affiliation in April, so the it was added in the revised manuscript (lines 16-17, page 1).

Re: Spectrum00391-24R1 (Functional and structural analysis of IMP-27 metallo- β -lactamase: Evolution of IMP-type enzymes to overcome Zn(II) deprivation)

Dear Dr. Akiko Shimizu-Ibuka:

Your manuscript has been accepted, and I am forwarding it to the ASM production staff for publication. Your paper will first be checked to make sure all elements meet the technical requirements. ASM staff will contact you if anything needs to be revised before copyediting and production can begin. Otherwise, you will be notified when your proofs are ready to be viewed.

Sincerely,
Pablo Power
Editor
Microbiology Spectrum